# Adversarial Environment Generation for Learning to Navigate the Web

## Abstract

Learning to autonomously navigate the web is a difficult sequential decision-making task. The state and action spaces are large and combinatorial in nature, and successful navigation may require traversing several partially-observed pages. One of the bottlenecks of training web navigation agents is providing a learnable curriculum of training environments that can cover the large variety of real-world websites. Therefore, we propose using Adversarial Environment Generation (AEG) to generate challenging web environments in which to train reinforcement learning (RL) agents. We introduce a new benchmarking environment, gMiniWoB, which enables an RL adversary to use compositional primitives to learn to generate complex websites. To train the adversary, we present a new decoder-like architecture that can directly control the difficulty of the environment, and a new training technique Flexible b-PAIRED. Flexible b-PAIRED jointly trains the adversary and a population of navigator agents and incentivizes the adversary to generate "just-the-right-challenge" environments by simultaneously learning two policies encoded in the adversary's architecture. First, for its environment complexity choice (difficulty budget), the adversary is rewarded with the performance of the best-performing agent in the population. Second, for selecting the design elements the adversary learns to maximize the regret using the difference in capabilities of navigator agents in population (flexible regret). The results show that the navigator agent trained with Flexible b-PAIRED generalizes to new environments, significantly outperforms competitive automatic curriculum generation baselines—including a state-of-the-art RL web navigation approach and prior methods for minimax regret AEG—on a set of challenging unseen test environments that are order of magnitude more complex than the previous benchmarks. The navigator agent achieves more than 75% success rate on all tasks, yielding 4x higher success rate that the strongest baseline.

## 1 Introduction

Autonomous web navigation agents that complete tedious, digital tasks, such a booking a flight or filling out forms, have a potential to significantly improve user experience and systems' accessibility. The agents could enable a user to issue requests such as, "Buy me a plane ticket to Los Angeles leaving on Friday", and have the agent automatically handle the details of completing these tasks. However, the complexity and diversity of real-world websites make this a formidable challenge.

General *web navigation form-filling tasks* such as these require an agent to navigate through a set of web pages, matching user's information to the appropriate elements on a web page. This is a highly challenging decision-making problem for several reasons. First, the observation space is large, and partially-observable, consisting of a single web page in the flow of several web pages (e.g. the payment information page is only one part of a shopping task). Web pages are represented using the Document Object Model (DOM), a tree of web elements with hundreds of nodes. Second, actions are all possible combination of the web elements (fill-in boxes, drop-downs, click on the buttons) and their possible values. For example, the drop-down selection actions are only appropriate if there there is a drop-down menu present. Even if the agent is able to navigate the site to arrive at the correct page, and eventually select the correct element (e.g. the 'departure' field for booking a flight), there are many possible values it can insert (e.g. all user input). Therefore, the action space is discrete and prohibitively large, with only a valid set of actions changing with the context. Finally, the same task, such as booking a flight, results in a very different experience and workflow depending on the

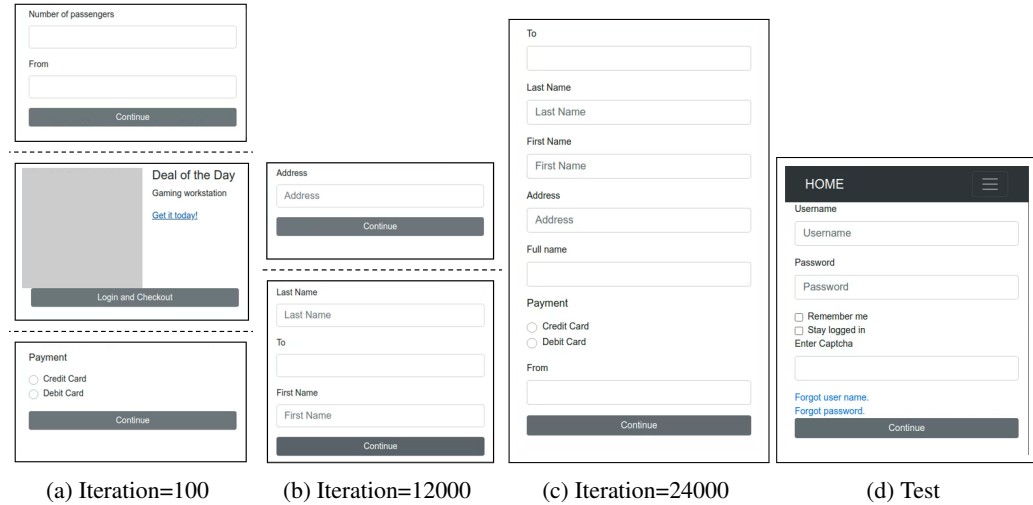

|  |  |  |  |
|---|---|---|---|
| (a) Iteration=100 | (b) Iteration=12000 | (c) Iteration=24000 | (d) Test |

Figure 1: Samples of generated web pages from selected websites taken from different snapshots of the training (a-c) and unseen test "Login" website (d). Over time, the number of pages in a website decreases but the density of elements in a page increases with more task-oriented elements. See Appendix A.11 for more samples.

website. The agent must be able to adapt and operate in the new environment to complete the task. Therefore, the reinforcement learning (RL) agents should be capable of zero-shot generalization to new environments.

Prior work made significant strides toward learning web navigation on a single website, yet the existing methods do not scale. Behavior cloning from expert demonstrations (Shi et al., 2017; Liu et al., 2018) shows promising results, however, it requires a number of demonstrations for every single website. RL agent trained using synthetic demonstrations created with a generative model Gur et al. (2019) improves the performance. Yet, the method still requires training a separate policy for every single website requiring tens of thousands of interactions with every website. Lastly, the existing benchmarks (Shi et al., 2017; Liu et al., 2018) have limited complexity. Their DOM trees are fixed and considerably smaller than real websites.

We aim to train RL agents to solve *web navigation form-filling tasks*; by correctly entering relevant information into *unknown* websites. Successful generalization to new websites requires training an agent on a large distribution of possible tasks and environments. The question is how to create a distribution that will not only cover most realistic tasks, but can be presented in a curriculum that is learnable by the agent. Manually designing a pre-defined curriculum of hand-built websites is tedious, and intractable. Another option would be to apply domain randomization (DR) (as in e.g. Jakobi (1997); Sadeghi & Levine (2016); Tobin et al. (2017)) to randomize parameters of websites, or automatically increase some parameter controlling the difficulty over time (as in Gur et al. (2019)). However, all these approaches are likely to fail to cover important test cases, and cannot tailor the difficulty of the parameter configuration to the current ability of the agent.

Adversarial Environment Generation (AEG) trains a learning adversary to automatically generate a curriculum of training environments, enabling both increased complexity of training environments, and generalization to new, unforeseen test environments. However, if we naively train a minimax adversary—i.e. an adversary that seeks to minimize the performance of the learning agent—the adversary is motivated to create the hardest possible website, preventing learning. Instead, PAIRED (Protagonist Antagonist Induced Regret Environment Design) (Dennis et al., 2020), trains the adversary to maximize the *regret*, estimated as a difference between two navigation agents (protagonist and antagonist). While PAIRED shows exciting results, without an explicit feedback on how skillful antagonist is and mechanism to control the difficulty of the environment, the method is susceptible to local minima, and has hard time learning in the complex environments when the regret is zero.

We present Flexible b-PAIRED, which builds on PAIRED framework, and jointly trains the adversarial RL agent (adversary) and a population of navigator agents. Flexible b-PAIRED adversary learns to present *"just-the-right-challenge"* to the navigation agents. We enable Flexible b-PAIRED adversary to tailor the environment difficulty to the ability of the best performing agent by introducing an explicit difficulty *budgeting* mechanism, and a novel multi-objective loss function. The budgeting mechanism gives the adversary the direct control of the difficulty of the generated environment. The

adversary training simultaneously optimizes for an objective that ties in adversary difficulty budget with the navigator agent's performance (observed expected return), and the population-based regret similar to PAIRED. Lastly, to enable AEG web-design, we present a new benchmarking environment, gMiniWoB, and a web-design adversary architecture. gMiniWoB enables an adversary to construct websites of *increasing complexity* out of common design primitives such as *navigation bars*, *product carousels*, *item decks*, *web forms*, and *item carts*. The evaluation environments in gMiniWob are order of magnitude more complex than miniWob (Shi et al., 2017). The adversary architecture is a LSTM-based decoder, seeded with a random seed. It first selects number of web pages. Then, at each step of an open loop, the adversary either emits a design element and its placement, or opts to skip an element and save design budget. The adversary's used difficulty budget is a log-likelihood of joint probability of not adding design elements.

This paper makes the following contributions: i) A new benchmarking environment, gMiniWoB, which empowers the use of AEG for web navigation, by enabling the construction of websites out of compositional design primitives; ii) The Flexible b-PAIRED algorithm, which computes a more stable estimate of regret and directly incentivizes the adversary to tailor the complexity of the generated environment to the performance of the best-performing agent; iii) web navigation adversary decoder architecture, and iv) empirical results demonstrating that Flexible b-PAIRED generates a curriculum of increasingly challenging websites, and produces agents that can successfully generalize to navigating complex, unseen sites at test time. Flexible b-PAIRED approach significantly outperforms prior work on minimax regret AEG (Dennis et al., 2020), as well as a state-of-the-art approach for using RL to train web navigation agents (Gur et al., 2019), resulting in agents that complete the most difficult tasks with more than 75% success rate, 4x improvement over the strongest baseline. We are releasing gMiniWoB in open-source in the hopes of enabling further progress on this problem. We hope that this work will provide a meaningful way to make progress on the exceptionally challenging problem of learning to navigate the web, and will be of interest to the wider RL research community for auto-curriculum design in complex and compositional environments.

## 2 RELATED WORK

**Web navigation benchmarks and tasks:** Prior work on training agents to navigate the web introduced the MiniWoB (Shi et al., 2017) and MiniWoB++ (Liu et al., 2018) environments, a fixed set of manually curated toy websites, but relied on obtaining expert demonstrations for each website, which cannot scale effectively to cover the large variety of real-world websites, and cannot adapt to changing websites. Further, these methods failed to solve complex web navigation tasks such as flight booking or social media interaction (Gur et al., 2019). Gur et al. (2019) take a step farther by training an RL agent to solve complex web navigation tasks using a scheduled curriculum. The curriculum linearly increases a parameter $p$, in which $1 - p$ controls the number of web elements that are solved by querying an oracle policy, which is obtained via expert data. This work differs in several ways. First, we introduce a new framework, gMiniWoB, that allows generating complex websites on-the-fly with tunable difficulty levels. Additionally, we do not rely on any expert demonstrations to augment sparse rewards. We use AEG to automatically learn to generate a curriculum of web navigation tasks that are tailored to the current skill level of the agent. Next, we make no assumption on the availability of any website while they assume websites are given *a priori*. Lastly, our web navigation agents generalize to unseen environments.

**Goal Generation:** Florensa et al. (2018) trains a Generative Adversarial Network (GAN) for generating a curriculum of goals with fixed environment dynamics. A generator is trained to output new goals and the discriminator is trained to predict if the goal is achievable. The generator is bootstrapped from sample goals that the initial agent is able to reach in the environment. It is tested on simple navigation tasks with the same environments. In contrast, we train an adversary that generates a curriculum of *environments*, including goals, starting with an empty environment in which bootstrapping a generator network from sample episodes is not possible. We test on unseen environments with more complicated and high dimensional state and action spaces.

**Adversarial Environment Generation:** Multi-agent training can be an effective method for automatically generating a curriculum of RL tasks (e.g. Leibo et al. (2019); Matiisen et al. (2019); Graves et al. (2017); Portelas et al. (2020)). For example, Asymmetric Self Play (ASP) (Sukhbaatar et al., 2017) trains two agents, in which the second agent must learn to repeat the actions taken by the first, demonstrator agent. Both agents play in the same, fixed environment. In contrast, we use a third agent to learn to generate challenging new environments. POET (Wang et al., 2019; 2020) is

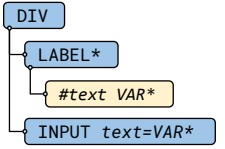 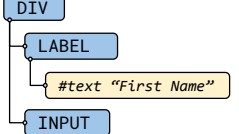 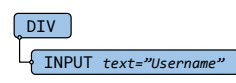

(a) An **underspecified** DOM tree template. The text box is always included, its text and label element are variables.

(b) A **fully specified** DOM primitive where a label is created and its text is assigned.

(c) A **fully specified** DOM primitive where only the inner text within the text box is assigned.

Figure 2: An example underspecified DOM tree template (a) and its instantiations (b,c) with different values. (*) indicates a variable; either an element or one of its attributes. (c) is used in Page 1 and (b) is used in Page 2 in Figure 3.

an AEG technique which uses a population of adversaries to generate the terrain a 2D walker agent must learn to navigate. To create a curriculum, POET requires generating many new environments, testing all agents within each one, and discarding environments based on a manually chosen a reward threshold, which wastes a significant amount of computation. Campero et al. (2020) use a teacher to propose navigation tasks; the teacher's reward is based on whether the agent takes more steps than a threshold, a hyperparameter that is linearly increased over the course of training.

Most closely related to our work is PAIRED (Dennis et al., 2020), which is an AEG method for training agents with minimal regret that works by constraining the environment-generating adversary using the performance of a second agent. However, PAIRED only demonstrated results on simple gridworld environments, and did not expand to the type of complex, high-dimensional state-action space required for web navigation. We improve on PAIRED using a more flexible estimate of the regret, as well as a budget mechanism, and show that this significantly improves performance.

**RL with Autoregressive Models:** Keneshloo et al. (2020) outlines training sequence-to-sequence (seq2seq) models with RL algorithms. Previous models first pretrained a seq2seq model with ground-truth inputs and outputs and then finetuned with RL using different reward functions such as BLEU score. In this work, we propose a decoder-like autoregressive adversary model that is trained without any ground-truth data. The model is fed its own predictions from previous time steps and updated using a novel adversarial objective.

## 3 BACKGROUND ON WEB NAVIGATION PROBLEM

Following previous work (Shi et al., 2017; Gur et al., 2019; Liu et al., 2018), we formulate web navigation as a sequential decision making problem where we train an agent, parameterized by a network $\pi(a_t|s_t; \Theta_i)$, that maps an input state $s_t$ to output action distribution to maximize the cumulative discounted reward, .i.e., $O = \sum_{t=0}^{T} \gamma^t r_t$ where $r_t$ is the reward at time step $t$, $\gamma$ is a discount factor, and $T$ is the length of an episode. We use the web page and user instruction as the input state. The web page is dynamically updated at each time step, while the instruction is fixed at the beginning of an episode. We represent web pages using Document Object Model (DOM), a tree of elements in a page, where each element is denoted by a set of (attribute, value) pairs and an array of features (such as spatial coordinates). Instructions are given as a set of fields where each field is a (key, value) pair. Keys are fixed for each task and values dynamically change based on user input.

Each action is represented as a tuple (element, field) that denotes acting on the element using the field as an input; i.e. typing the value of the field into the element. Agents receive a task success reward (1.0 or -1.0) at the end of each episode, a potential-based reward when the value of an element in the page is updated, and a small penalty each timestep to encourage efficient navigation. As an example, consider a flight booking task where the agent is given an instruction {`"Departure Date"`: `"Friday"`, `Destination Airport`: `"Los Angeles (LAX)"`}. The agent first picks a field (e.g. destination airport) and finds the corresponding text box in the page; then the corresponding value ("Los Angeles (LAX)") typed in to the text box. If this value is correct, the agent receives a positive reward of $1/2$ where 2 is the number of fields in the instruction.

## 4 METHODS FOR LEARNING TO DESIGN WEB ENVIRONMENTS

This Section presents the generative MiniWob environment (Section 4.1), the adversary neural network architecture (Section 4.2) and the adversary training procedure in Section 4.3.

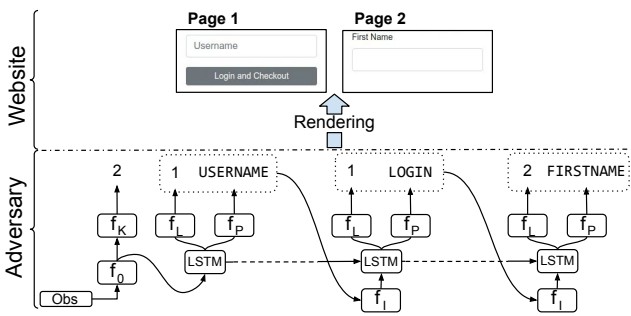

Figure 3: A sample rollout of the adversary for compositional environment generation for web navigation problem. An initial observation (Obs) is given at the beginning of the rollout. $f_0$, $f_K$, $f_L$, $f_P$, and $f_I$ denote networks for encoding initial observation, generating number of pages, page indices,[1] primitives, and encoding LSTM inputs, respectively.

## 4.1 GENERATIVE MINIWOB (GMINIWOB) ENVIRONMENT

Generative MiniWoB (gMiniWoB) creates website environments for form-filling tasks, which consist of a linked-list of webpages, $\mathcal{E}_w = [\mathcal{W}_1, \cdots, \mathcal{W}_K]$. Each webpage, $\mathcal{W}_i$, is a DOM tree that contains a number of elements, such as fill-in boxes, drop downs, and buttons. To create a new environment $\mathcal{E}_w$, gMiniWoB starts with an empty website that is gradually populated by new pages, $\mathcal{W}_i$. A subset of elements are also augmented with events that enable page transitions. For example, an "on-click" event on a Submit button on a page $\mathcal{W}_i$, will link to a page $\mathcal{W}_{i+1}$.

We formulate the website design as combining a set of *primitive DOM sub-trees* that are general enough to create complex websites but also facilitate controllable generation. The order in which the primitives are combined also defines how the web page will be rendered as well. Let's assume that we are given a list of DOM tree primitives $\mathcal{T}$ and an empty web page $\mathcal{W} = (\mathcal{S}, \mathcal{C})$ where $\mathcal{S}$ is a single root node of the DOM tree and $\mathcal{C}$ is an ordered list of subtrees rooted at $\mathcal{S}$ which is initially empty. By repetitively sampling new primitives from $\mathcal{T}$ and appending them to $\mathcal{C}$, we create a new page, $\mathcal{W}$, which follows the order of primitives in $\mathcal{C}$ when rendered (see Figure 3).

We first create a set of *underspecified DOM tree templates*, a sub-tree with certain elements and attributes are replaced with variables. Assigning values to variables in a template fully specifies a DOM tree primitive that is placed in a subtree $\mathcal{C}$ to create a new web page, $\mathcal{W}$ (Algorithm 2). For example, an input template (Figure 2a) as a variable label and text box with a common parent. There two ways to assign values, either by picking the label element and assigning a value to its text attribute (Figure 2b), assigning a value to the inner text of the text box and ignoring the label element (Figure 2c).

**Website Design Primitives:** gMiniWoB implements 40 different design primitives from 11 different underspecified DOM templates. The primitives are widely used across the web and include 'navigation bars', 'product carousels', 'item decks', 'web forms', 'item carts', 'dropdowns', etc. Every primitive includes at least one actionable element that changes the DOM structure when the agent interacts with it. Each primitive belongs to one of the two categories based on their use in the reward computation: (i) **Active primitives (used)**, and (ii) **Passive primitives (not used)**. 26 of the 40 primitives are active, while the rest are passive. When an active primitive is added to a web page, the instruction automatically grows as well. For example, adding 'First Name' text box in Figure 2c also adds a new *"firstname"* field into user instruction. Expanding the instruction set makes active primitives more complicated to learn, while the passive primitives mostly serve as noise. However, real websites contain many distracting elements (passive primitives), so it is important for agents to learn to ignore them. Appendix A.9 details all the design primitives used, and Appendix A.10 shows the websites in the test set.

---

[1] For simplicity of illustration, we show an example generation process where primitives are generated in an increasing order of the page indices; however, in our formulation (see Section 4.2 for details), the page indices corresponding to consecutive LSTM timesteps do not necessarily increase monotonically.

## 4.2 ADVERSERIAL ENVIRONMENT DECODER ARCHITECTURE

We present an adversary decoder policy for the compositional environment generation problem where the goal is to place a set of design primitives to a set of locations. Adversary generates a (website) environment, $\mathcal{E}_w = [\mathcal{W}_1, \cdots, \mathcal{W}_K]$. We assume fixed maximum number of pages $K$, although to control complexity, we allow pages and subtress to be empty.

We parametrize the adversary with a policy $\pi_E(a^A|o^A)$ such that

$$\pi_E(a^A|o^A) = \pi_{\mathcal{E}_w}(k|K) \prod_{i=0}^{N} \pi(a_i, b_i|a_{0\cdots i-1}, b_{0\cdots i-1}, k) \tag{1}$$

where $N$ is an upper limit on the number of outputs (total number of primitives in the environment $\mathcal{E}_w$), $K$ is an upper limit on the number of web pages, $a_i$ is a design primitive. We augment the primitive design actions described in Section 4.1 with a special SKIP action that does nothing when executed by the renderer. This allows the adversary to control the number of primitives added. $b_i$ is a web page index of where the primitive $a_i$ should be placed in on the page. Observation $o^A$ is an initial observation. The adversary first samples the number of locations $k$ from a parametrized Categorical distribution $Cat(0, K)$. Conditioned on $o^A$, it executes an autoregressive model to generate a set of primitives and their corresponding locations within $[0, \cdots, k]$. $a^A$ is the resulting environment totalling at most $N$ primitives placed over $k$ pages.

The initial observation $o^A$ is sampled from the standard normal distribution, to allow the adversary to diversify its design distribution. This observation is encoded with a feed forward network $h_0 = f_0(o^A)$ and $h_0$ is passed to another network $f_K$ that outputs a distribution over number of empty pages. The same hidden state $h_0$ is passed to an LSTM network as the initial input vector and output of the LSTM is used by two independent networks $f_P$ and $f_L$ to (i) learn a distribution over design primitives and (ii) learn a distribution over locations, respectively. We sample an action (a joint primitive and location pair) from these distributions and they are encoded by another network $f_I$ into a hidden state which is used as the input to the LSTM at the next step. After $N$ steps, sampled design actions are sent to a renderer module which generates the environment (Figure 3).

Note that the adversary is domain independent, and creates a generic compositional task environment with $K$ linked sub-environments, each containing sub tasks sampled from the design primitives. Since the renderer interprets the design decisions and builds the environments for the navigator agents to use, the adversary architecture can be used in other domains without modification.

## 4.3 ADVERSARY TRAINING

We train both the adversary and navigation agents with reinforcement learning. At every step $t$ of the training, the adversary generates an environment $\mathcal{E}_w(t)$. The web navigation training initializes population of navigation agents $\mathcal{A} = \{A_i, i = 1, \cdots, n_a\}$. The agents $A_i \in \mathcal{A}$ collect $M$ trajectories $\tau_{i,j}$ with returns $R_{i,j}$. $R_{i,j}$ is a discounted cumulative reward that agent $A_i$ observers by navigating the environment, $\mathcal{E}_w$, and resulting in trajectory $\tau_{i,j} \sim \pi_{A_i}$. The navigation agents use the standard task related reward described in Section 3 for training and out-of-box A2C with entropy (Mnih et al., 2016). To train the adversary, we present a new loss function which augments A2C with entropy with a custom loss function that encourages the adversary to control the complexity of the environment, by presenting "just-the-right" challenge for the agents in $\mathcal{A}$ (see Algorithm 1).

### 4.3.1 LOSS FUNCTIONS

Let $R_{i,j}(t)$ be observed returns (cumulative discounted reward) of an agent $A_i(t) \in \mathcal{A}$ while sampling $j^{\text{th}}$ trajectory during training iteration $t$. Let $R^A$ and $R^P$ be maximum and average expected returns at the iteration $t$,

$$R^A = \max_i \mathbb{E}(R_i), \ \ R^P = \mathbb{E}[\mathbb{E}[R_i]], \ \ \mathbb{E}[R_i] = \frac{1}{M} \sum_j R_{i,j}, \ i = 1, \cdots, M \tag{2}$$

Then, the adversary loss function consists of two terms,

$$\mathcal{J}(\theta|\mathcal{A}, \mathcal{E}_w) = \mathcal{J}_{\text{rl}}(\theta \,|\, \text{REGRET}(\mathcal{A}|\mathcal{E}_w)) + \alpha * \mathcal{J}_{\text{budget}}(\theta \,|\, \mathcal{A}, \mathcal{E}_w) \tag{3}$$

---

**Algorithm 1** b-Flexible PAIRED training. Joint training of the adversary and navigation agents.
_________________________________________________________________________________
1: **Input:**$\mathcal{A}$: Initialize the agents independently
2: **for** all training iterations **do**
3:     $\mathcal{E}_w \longleftarrow$ Run the adversary $\pi_E$ to generate a new website
4:     **for** $i = 1, \cdots, n_a$ **do**
5:         $R_i \longleftarrow 0$
6:         **for** $j = 1, \cdots, M$ **do**
7:             $R_{i,j} \longleftarrow$ Run agent $A_i \in \mathcal{A}$ in the environment $\mathcal{E}_w$ and collect rewards.
8:             $R_i \longleftarrow R_i + \frac{R_{i,j}}{M}$                           ▷ Expected return for agent $\mathcal{A}_i$.
9:         **end for**
10:    **end for**
11:    $R^A \longleftarrow \max_i R_i, R^P \longleftarrow \mathbb{E}[R_i]$    ▷ Maximum and mean expected return from the agents.
12:    REGRET$(\mathcal{A}|\mathcal{E}_w) \longleftarrow R^A - R^P$                  ▷ Compute regret as in Equation 4.
13:    Update adversary using Equation 3 and REGRET$(\mathcal{A}|\mathcal{E}_w)$ as the reward.  ▷ Train adversary.
14:    Update parameters of $\forall A_i \in \mathcal{A}$ using $R_i$ returns in A2C.       ▷ Train navigation agents.
15: **end for**
_________________________________________________________________________________

where $\alpha$ is a balancing factor between the two losses. $\mathcal{J}_{rl}$ is standard A2C loss with cross-entropy regularizer added to encourage adversary's exploration. The reward function for the $\mathcal{J}_{rl}$ is regret, estimated as the difference between expected performance of the best and average agents:

$$\text{REGRET}(\mathcal{A}|\mathcal{E}_w) = R^A - R^P. \tag{4}$$

The second loss term is budget loss,

$$\mathcal{J}_{\text{budget}}(\theta \mid \mathcal{A}, \mathcal{E}_w) = R^A * \sum_{i=1}^{N} \log \pi_\theta(a_i = \text{SKIP}|a_{0\cdots i-1}, b_{0,\cdots,i-1}). \tag{5}$$

We use an environment difficulty objective to bind the adversary's design budget to the performance of the best agent. We approximate the effective budget of the adversary as the expected number of non-SKIP actions over $N$ time steps and update this budget according to whether the agents are learning. This objective encourages the adversary to use less budget (more SKIP actions) when the agents are not yet learning (i.e., $R^A$ is negative or low); it encourages the adversary to use more budget (less SKIP actions) when the navigator agents are performing well and collecting positive rewards in the environment. On the flip side, when the agents are collecting negative reward, the adversary is encouraged to decrease the budget and sample less design elements (and more SKIP actions).

### 4.3.2 LOSS FUNCTIONS DISCUSSION

Budget loss provides training signal for the adversary when the regret reward is sparse, which happens when all agents are performing very similarly, and it acts to encourage the adversary to decrease the difficulty of the environment. Consider the scenario where agents are placed on the home page of a shopping website where there are many possible elements, but only a single button that takes them to their account page. During exploration, agents mostly collect negative rewards for taking incorrect actions, bounded to a very narrow interval (as there is only a single optimal action). In this case, the regret is very small and sparse, which hinders the adversary's ability to design environments at an appropriate difficulty for agents to learn. We approximate the environment difficulty and the effective budget of the adversary as the expected number of non-SKIP actions over $N$ created elements, and update this budget according to whether the agents are learning.

The regret presented in Equation 4 contributes two subtle, but important changes over the prior work. First, the presented regret does not make a distinction between antagonist and protagonist agents, and instead annotates the best performing agent as the antagonist. As long as any agent has a higher performance than the other agent, the objective will continue to improve the weakest agent. During that time, the other agents continue learning, and therefore provide a stronger maximum performance against which we measure the regret. This is why we call it *flexible*. Second, the estimates for the protagonist returns are smoothed and computed as a best mean return, instead of

maximum of a pre-selected agent. While this might be further underestimating the regret, it provides a more stable estimate.

Note that when the adversary creates non-trivial environments, REGRET is close to zero and the budget loss in Equation 5 dominates the RL loss. However, when the adversary creates trivial websites, the RL loss and regret encourage the adversary to explore towards environments that promote variation in performance. Those are less trivial, hence it pushes the adversary towards increasing the difficulty.

# 5 Evaluations

## 5.1 Evaluation setup

We evaluate our models on a variety of web environments implemented in gMiniWoB as well as MiniWoB frameworks (Shi et al., 2017; Liu et al., 2018). We implemented several challenging websites with varying difficulty levels using the same set of design primitives in gMiniWoB. These environments include 'Login', 'Enter Address', 'Flight Booking', 'Enter Payment', and 'Shopping' websites, where the agents need to enter text or select information in the website while navigating between pages. Each environment comes with 4 different difficulty levels by gradually adding more primitives to websites. These environments are never explicitly presented to agents during training, so performance in them measures how well agents can generalize to unseen websites at test time.

**Agent architecture:** Following Gur et al. (2019), we utilize an LSTM based DOM tree encoder and a feed forward network to encode profile fields. The navigator agent policy outputs a joint distribution over elements and fields by measuring pairwise similarities between element encodings and profile fields. We compute the state-value by using the marginal distribution of elements as attention weights over element encodings and passing the context vector through a FF network. Web navigation agents are trained with an actor-critic algorithm (Liu et al., 2018). We train the LSTM-based adversary network using Flexible PAIRED and Flexible b-PAIRED with policy gradient (See Appendix A.12 for more details on adversary policy network).

**Baselines:** We benchmark PAIRED, Flexible PAIRED, and Flexible b-PAIRED against two additional baselines. First, a Domain Randomization (DR) agent, which we implement using a similar approach as Dennis et al. (2020). We first sample the number of empty pages $k$ from a uniform distribution $U[0, K]$. Next, we randomly sample a primitive (including SKIP), and a page from $U[0, k]$ for $N$ steps. Second, a Curriculum Learning (CL) approach, which adapts the scheduled curriculum idea of Gur et al. (2019) to zero-shot environment generation where we are not given a specific website but a set of design primitives. We randomly sample each primitive w.r.t. a probability $p$ where $p$ is initialized with a small number and scheduled to reach 1.0 during training.

## 5.2 Results

We first compare the original PAIRED algorithm (which used separate antagonist and protagonist agents) to the proposed Flexible PAIRED algorithm that annotates the best performing agent as the antagonist. Flexible PAIRED considerably improves upon PAIRED, which fails to learn in this environment (Figure 4). One reason is that when agents are separate and have very similar rewards, especially early during training, the regret becomes very small. This uninformative signal makes it difficult for the adversary to learn. On the other hand, Flexible PAIRED computes a consistently positive regret signal, which more clearly indicates to the adversary which environments are challenging, but still feasible. The further ablation studies show that adding budget improves performance for both flexible, and original PAIRED method (see Appendix A.7 for MiniWoB studies and Appendix A.8 for budget weight ablation).

**Comparison on test environments:** We evaluate the performance of the proposed models and baselines on task success rate computed across test environments with different difficulty levels. Flexible b-PAIRED outperforms Flexible PAIRED indicating the budget objective significantly improves performance (Figure 5). Further, both techniques significantly outperform the baseline models on all tasks, with Flexible b-PAIRED effectively reaching more than 75% task success across all difficulty levels. Even as the complexity of the environments continues to increase (see Section 5.2), Flexible b-PAIRED agents still perform consistently well without degrading performance. While CL outperforms Flexible PAIRED early in the training, its performance drops significantly due to ignoring

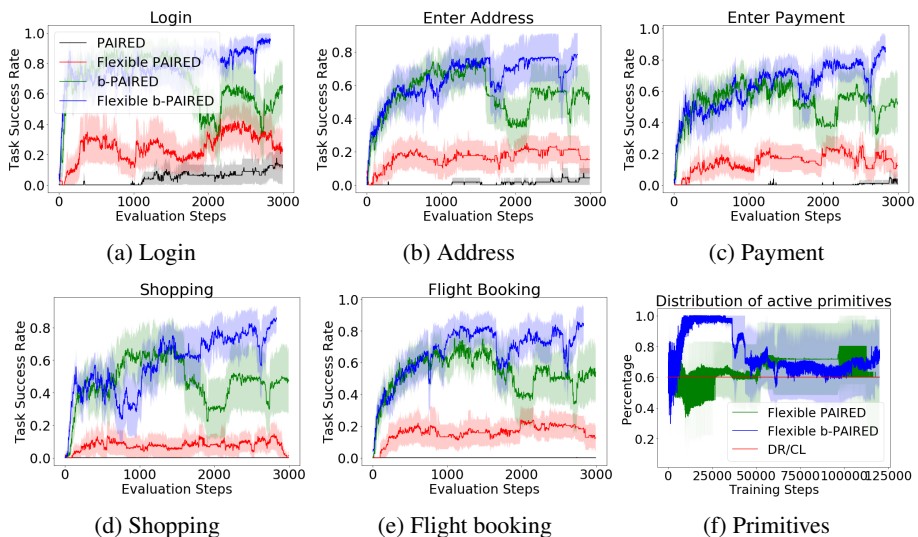

Figure 4: Comparison of PAIRED (Dennis et al., 2020) and Flexible PAIRED with and without budget enforcing; averaged over 4 difficulty levels. (f): Percentage of active primitives over training steps (see Appendix A.4 for more details).

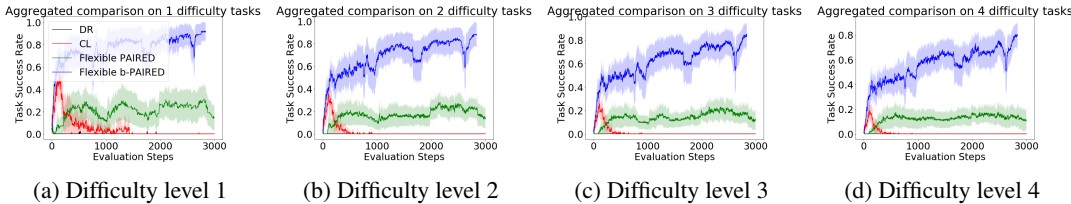

Figure 5: Aggregated task success rate comparison of Flexible b-PAIRED and baseline models on test environments with increasing difficulty levels. See Appendix A.6 for detailed results.

agents' skill level, and making environments that are too challenging for agents to complete. We also observe that Flexible b-PAIRED learns faster than Flexible PAIRED on all environments as Flexible b-PAIRED reacts to agents' performance faster than Flexible PAIRED (see Appendix A.6).

**Environments complexity:** While agent performance improves over time, we would like to know if they are presented with more challenging environments over training. We estimate the number of active and passive primitives generated as a measure of environment complexity. Learning a web page with more passive primitives is a relatively easier task than a page with more active primitives, because passive primitives either add noise and should ignored by the agents, or are used by agents only to navigate to another page. On the other hand, if there are more active primitives, not only will the size of the DOM tree increase but the number of profile fields will increase, making the matching between elements and profile more challenging. Flexible b-PAIRED starts around 60% random selection of primitives, and gradually generates more active primitives early on (Figure 4f). Although presented with more active primitives by Flexible b-PAIRED, agents are still able to improve thanks to Flexible b-PAIRED's ability to accurately tune the difficulty of the environments according to agents' skill. During training, more and more passive primitives are also introduced where number of active primitives also keeps increasing (see Appendix A.4). We also observe that the distribution of the primitives shifts later in the training to more complex and relevant primitives (see Appendix A.3).

## 6 CONCLUSION

This work presents a novel technique for Adversarial Environment Generation (AEG), which we show improves significantly over prior work. In addition, we apply AEG to the problem of web navigation, and provide an open-source environment that enables learning to design complex websites out of a set of compositional primitives. Our Flexible b-PAIRED method is able to generate a curriculum of increasingly complicated websites, and successfully trains agents which can navigate challenging, high-dimensional websites.

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

# A  APPENDIX

## A.1  Protagonist Antagonist Induced Regret Environment Design (PAIRED)

Adversarial Environment Generation (AEG) trains an adversary policy $\pi_E$ to design environments to minimize the performance of an agent's policy, $\pi_P$. Let $R_i^P = \sum_{t=1}^{T} \gamma^t r_t^P$ be the total reward received by the agent for trajectory $i$. In minimax AEG, the objective for the adversary is simply: $-R^P$. Thus, minimax adversaries are incentivized to create excessively difficult or impossible environments, which may not enable the agent to learn. Instead, PAIRED (Dennis et al., 2020) trains the adversary to maximize the agent's *regret*, which is defined as the difference between the agent's return and the return of the optimal policy, $R^* - R^P$. When the reward function includes an incentive to complete the task more efficiently (which is true in our case), the regret will be highest for easy tasks which could be completed in a few steps by the optimal policy, but which the current policy fails to complete. Therefore, an adversary that maximizes the regret will continue to propose easier tasks until the agent begins to solve them, making regret a desirable objective for AEG.

To estimate the regret, PAIRED introduces a third agent, the *antagonist* (with policy $\pi_A$), and constrains the adversary to only generate feasible environments which the antagonist can complete. When the adversary generates an environment $E$, both the protagonist and antagonist collect $M$ trajectories with returns $R_1^P, ..., R_M^P, R_1^A, ..., R_M^A$ in $E$. The regret is then estimated as:

$$\text{REGRET} = \max_i R_i^A - \frac{1}{M} \sum_{m=1}^{M} R_m^P \tag{6}$$

As Dennis et al. (2020) show, if the adversary and antagonist coordinate and reach a Nash equilibrium with the protagonist, then the protagonist will have learned to minimize the regret. However, in practice gradient-based multi-agent RL has no convergence guarantees, is highly non-stationary, and will often fail to converge (Mazumdar et al., 2019a;b). If the antagonist and adversary in PAIRED fail to coordinate, then PAIRED minimizes regret with respect to the antagonist's policy. In that case, the objective in Equation 6 only forces the protagonist to learn to be as good as the antagonist. If the antagonist fails to improve, or reaches a local optimum, then the adversary cannot continue to train the protagonist. In Section 4.3.1 we propose an improved objective which addresses this problem.

## A.2  Training flow

In Figure 6, we illustrate the high level workflow of the AEG with budget mechanism.

## A.3  Distribution of Primitives During Training

During training, the distribution of primitives become more skewed towards active primitives early on (as shown in Figure 4f), but as the environments get more challenging, more passive primitives are introduced as well (Figure 7). What we observe from the histograms in Figure 7 is that new primitives are slowly introduced while the ranking of the primitives is also slightly changed.

## A.4  Active and Passive Primitive Frequencies

In Figure 8, we present frequencies of active and passive primitives during training. With Flexible-bPAIRED, number of both active and passive primitives increase resulting in more complex websites.

## A.5  Creating fully-specified primitives from underspecified templates

In Algorithm 2, we outline the process for generating a new fully-specified primitive from a given underspecified DOM template.

---

**Algorithm 2** Generating a new fully-specified primitive from an underspecified primitive.

---

1: **Input:** $D = (D_n, D_e)$: An underspecified DOM template, a sub-tree with elements $D_n$ and edges $D_e$
2: **Input:** $V \subset D_n$: A list of elements that correspond to variables in $D_n$
3: **Input:** $A_{v,i}$: A list of variable attributes $A_{v,i}$ for an element $v \in D_n$
4: **for** $v \in V$ **do**                                                  ▷ Iterate over variable elements.
5:     Flip a coin. If it is heads, $D_n \longleftarrow D_n \setminus \{v\}$.          ▷ Add/remove a variable element.
6: **end for**
7: **for** $v \in D_n$ **do**                                          ▷ Iterate over non-variable elements.
8:     **for** $a \in A_{v,i}$ **do**                              ▷ Iterate over variable attributes for element $v$.
9:         Flip a coin. If it is heads, sample and assign a value for $a$.    ▷ Add/remove an attribute.
10:    **end for**
11:    If there is at least one variable attribute remaining for element $v$, $D_n \longleftarrow D_n \setminus \{v\}$.
12: **end for**

---

## A.6    DETAILED RESULTS ON TEST ENVIRONMENTS

We detail the aggregated results in Figure 5 and present performance of agents across tasks and difficulty levels (Figure 1). On the easiest level of tasks, CL achieves slightly lower performance than Flexible b-PAIRED early in the training while as the task difficulty increases, the gap becomes more apparent. We observe that the primitive distribution in Figure 7c and task success rate results are consistent in which late in the training, the adversary focuses more on the 'Flight Booking' related primitives and its performance still strongly increases.

## A.7    RESULTS ON MINIWOB ENVIRONMENTS

In Table 2, we present results on MiniWoB form-filling tasks and compare them to gMiniWoB test tasks. MiniWoB tasks are independent of gMiniWoB and they have completely unobserved DOM structures and labels. We load only the trained embedding layers from the final checkpoint as there is no element dependency on these DOMs. We show that we navigation agents trained with Flexible b-PAIRED are able to solve all MiniWoB tasks.

Compared to gMiniWoB tasks, they are much simpler where DOM and instruction sizes are up to 10 times smaller. They also have only a few input elements that the agent can interact with while in gMiniWoB there are 10s of input elements making the gMiniWoB a formidable benchmark. As an example, in the Shopping task, size of the state and action spaces reach 5550 (number of tokens for all attributes in all elements in a DOM) and 240 (total number of element and instruction pairs), respectively.

## A.8    COMPARISON OF $\alpha$ IN BUDGET WEIGHTING

In Figure 9, we plot results where Flexible b-PAIRED is trained with different $\alpha$ weights. For $\alpha = 0.25$, the performance drops substantially as the model gives more weight to the RL loss overall. In this work, we used $\alpha = 1.25$ for our main results.

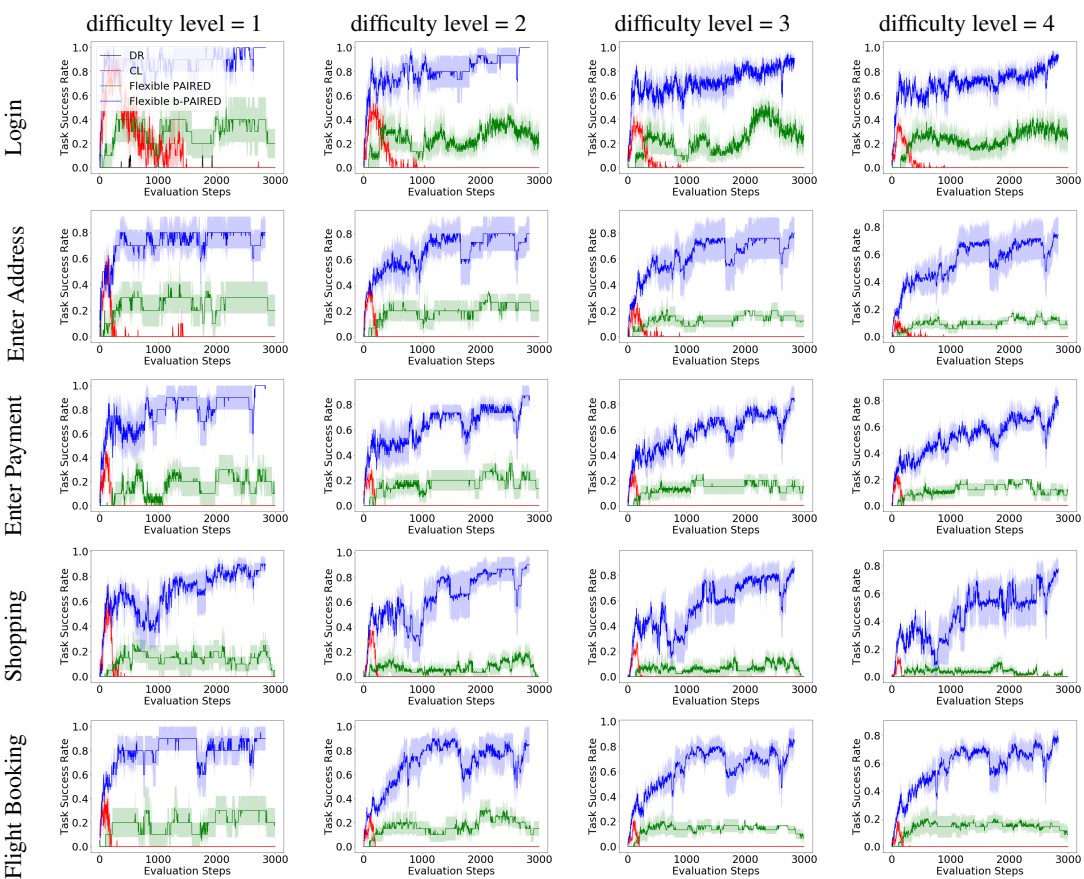

Table 1: Task success rate comparison of PAIRED and baseline models on test environments with increasing difficulty levels. From left to right, columns correspond to increasing difficulty. From top to bottom, rows correspond to different test environments.

| Task Name | Task Success | DOM Size (Inputs) | DOM Size | DOM Depth | Instruction Size |
|---|---|---|---|---|---|
| enter-password | 1.0 | 3 | 11 | 5 | 1 |
| enter-text | 1.0 | 2 | 6 | 4 | 1 |
| enter-text-dynamic | 1.0 | 2 | 6 | 4 | 1 |
| login (gMiniWoB) | 0.92 | 10 | 34 | 21 | 5 |
| address (gMiniWoB) | 0.74 | 10 | 38 | 22 | 7 |
| payment (gMiniWoB) | 0.78 | 13 | 49 | 28 | 5 |
| flight (gMiniWoB) | 0.79 | 16 | 60 | 33 | 7 |
| shopping (gMiniWoB) | 0.77 | 20 (40) | 111 (183) | 68 (111) | 12 |

Table 2: MiniWoB form-filling environment results compared to gMiniWoB test environments. Inputs correspond to elements that are interactable by agents. For shopping, we report maximum and total (in parentheses) across 3 pages for DOM related statistics.

## A.9 WEB ENVIRONMENT DESIGN PRIMITIVES

### Design Primitives and Their Descriptions

| Design Primitive | Design Template | Active/Passive | Description |
|---|---|---|---|
| addressline1 | input | active | Main address information |
| addressline2 | input | active | Secondary address information |
| cabin | multi-selection | active | Multiple cabin options |
| captcha | input | active | Captcha information |
| carousel | carousel | passive | Items with images in a carousel with previous and next buttons |
| cart | cart | passive | Items in a product cart with promo code information |
| cc | multi-selection | active | Multiple credit card type options |
| cccvv | input | active | Credit card CVV information |
| ccexpdate | input | active | Credit card expiration date information |
| ccnumber | input | active | Credit card number information |
| city | input | active | City address information |
| dealmedia | media | passive | Product media with image, label, and link |
| deck | deck | passive | Multiple product decks with image, label, and link |
| departureairport | input | active | Departure airport information |
| departuredate | input | active | Departure date information |
| destinationairport | input | active | Destination airport information |
| destinationdate | input | active | Destination date information |
| firstname | input | active | First name information |
| flighttype | multi-selection | active | Multiple flight type options |
| footer1 | footer | passive | Footer with links and information |
| forgotpassword | link | passive | Link with forgot password context |
| forgotusername | link | passive | Link with forgot username context |
| fullname | input | active | First and last name information |
| header | label | passive | Generic header |
| header_login | label | passive | Header for login form |
| header_select_items | label | passive | Header for item selection |
| inpgroup1 | input | passive | Generic input with default search context |
| lastname | input | active | Last name information |
| navbar | navigation bar | passive | A navigation bar with a menu |
| next_checkout | button | passive | Next button with checkout context |
| next_login | button | passive | Next button with login context |
| next_login_page | button | passive | Next button with login context |
| numberofpeople | multi-selection | active | Multiple number of people options |
| password | input | active | Password information |
| rememberme | selection | active | Checkbox with remember me context |
| state | input | active | State information |
| stayloggedin | selection | active | Checkbox with stay logged in context |
| submit | button | passive | Submit button |
| username | input | active | Username information |
| zipcode | input | active | Zipcode information |

In Table A.9, we present the list of design primitives, corresponding templates, types, and descriptions.

### A.10    LIST OF TEST ENVIRONMENTS

In Figure 11, we present screenshots of the testing environments with the hardest difficulty levels. While "Login", "Enter Address", "Enter Payment", and "Flight Booking" are single page environments, "Shopping" is a multi-page environment where an agent needs to first navigate the home page and then solve "Login" and "Enter Address" tasks.

### A.11    EXAMPLE WEB PAGE DESIGNS

In Figure 12, we present more screenshots of generated pages by the adversary from including multi-page websites. They cover a very broad spectrum of complexities and DOM tree structures. As an example, two web pages on the top right both have "City", "CVV", and "Address" elements but with different orders. This allows the web navigation agents to observe a website in multiple different ways for better generalization.

### A.12    IMPLEMENTATION DETAILS ON WEB NAVIGATION AND ADVERSARY NETWORKS

Following Gur et al. (2019), we design web navigation agent networks as DOM and profile encoders with pairwise similarity scoring. Each web navigation agent policy network has 104501 parameters.

In Figure 13, we detail the adversary network architecture for a single design action with the parameters used in this work. We use 100 dimensions for hidden vectors for all dense layers as well as the LSTM network. Every dense layer is stacked twice and tanh activation function is applied on the output of all non-final dense layers. Total number of parameters for the adversary policy network is 152461.

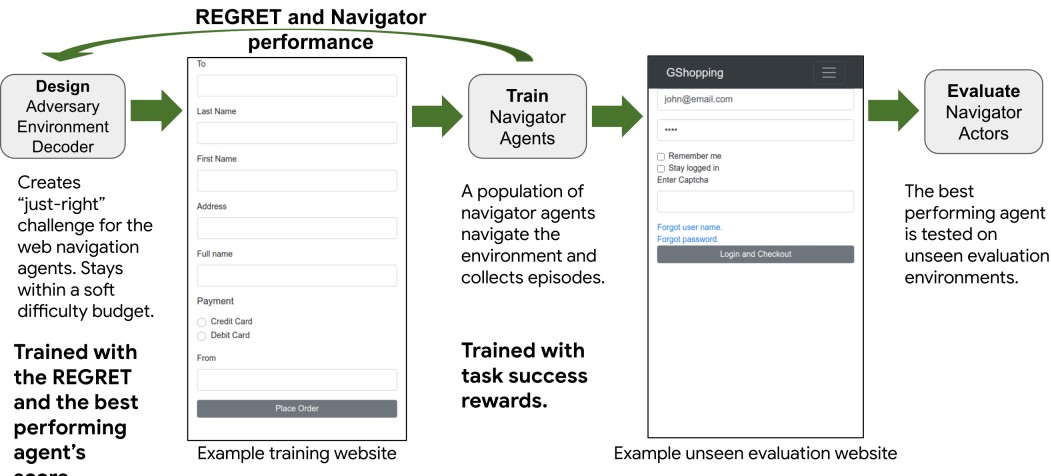

Figure 6: Training workflow. The adversary unrolls an environment, adding one element at the time of each page. That environment is passed on a population of navigation agents under training. The navigators collect several episodes in the environment, and collect their estimates returns. The weight of the navigator agents are updated w.r.t. their scores. And the adversary weights are updated w.r.t. regret estimate and performance score of the best performing agent.

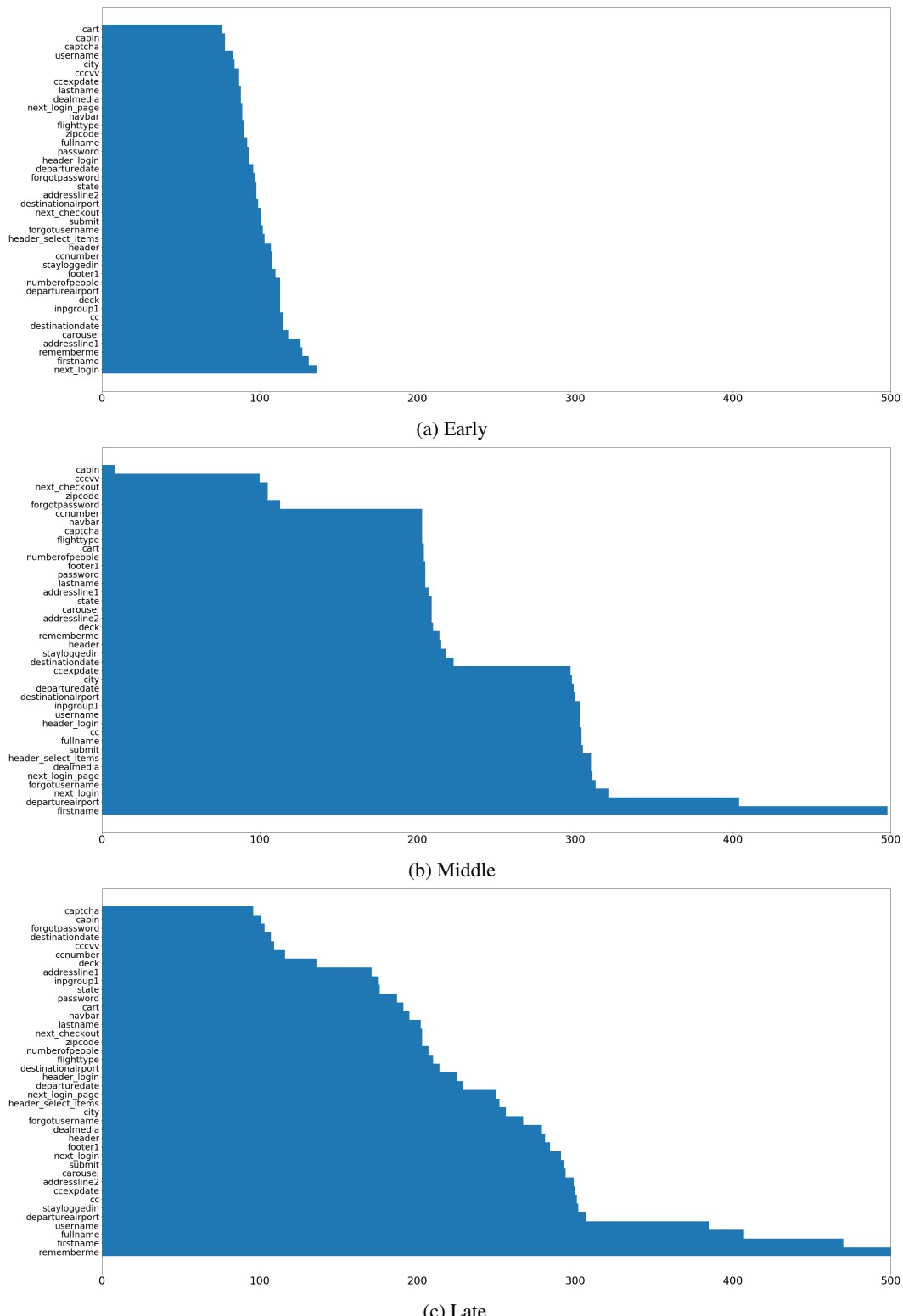

(a) Early

(b) Middle

(c) Late

Figure 7: Histograms of primitives from early, middle, and late snapshots of the training.

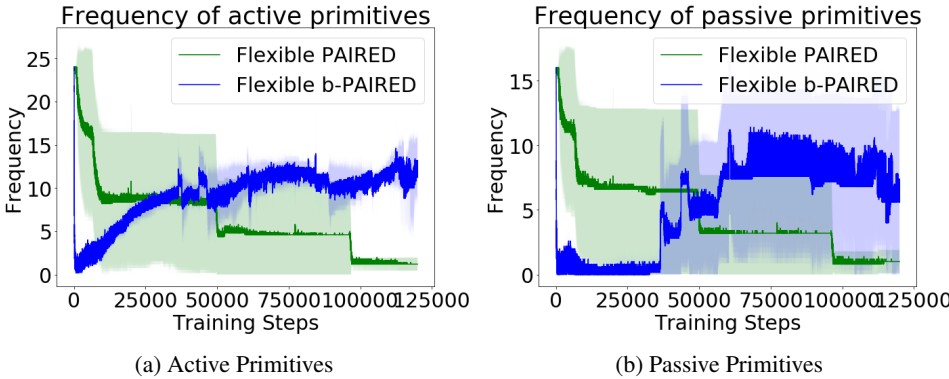

(a) Active Primitives

(b) Passive Primitives

Figure 8: Frequencies of active and passive primitives during training.

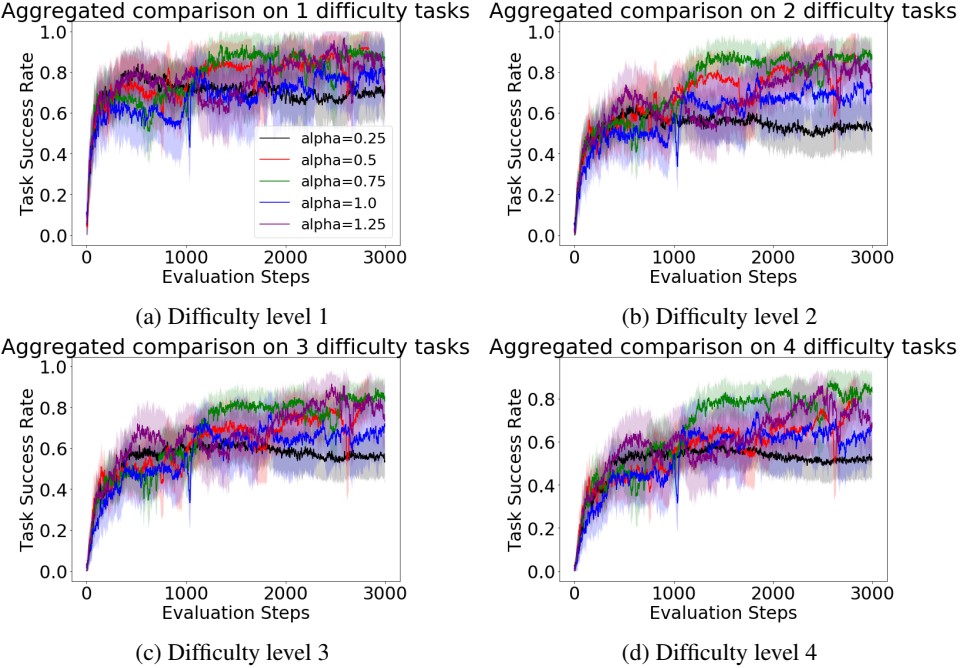

(a) Difficulty level 1

(b) Difficulty level 2

(c) Difficulty level 3

(d) Difficulty level 4

Figure 9: Aggregated task success rate comparison of Flexible b-PAIRED trained with different $\alpha$ weights.

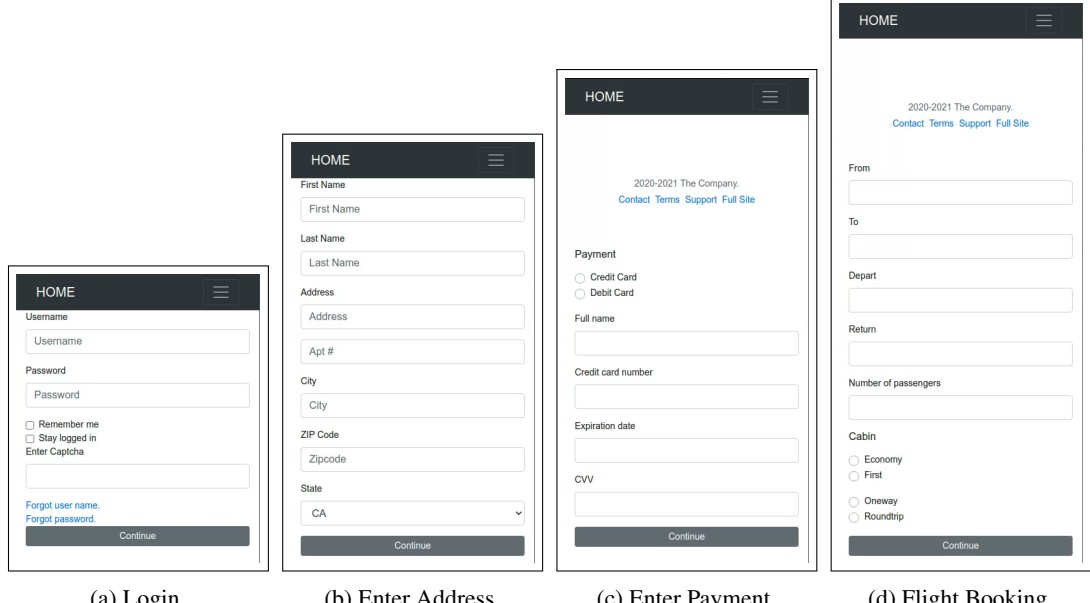

(a) Login       (b) Enter Address       (c) Enter Payment       (d) Flight Booking

Figure 10: Screenshots of single page test environments.

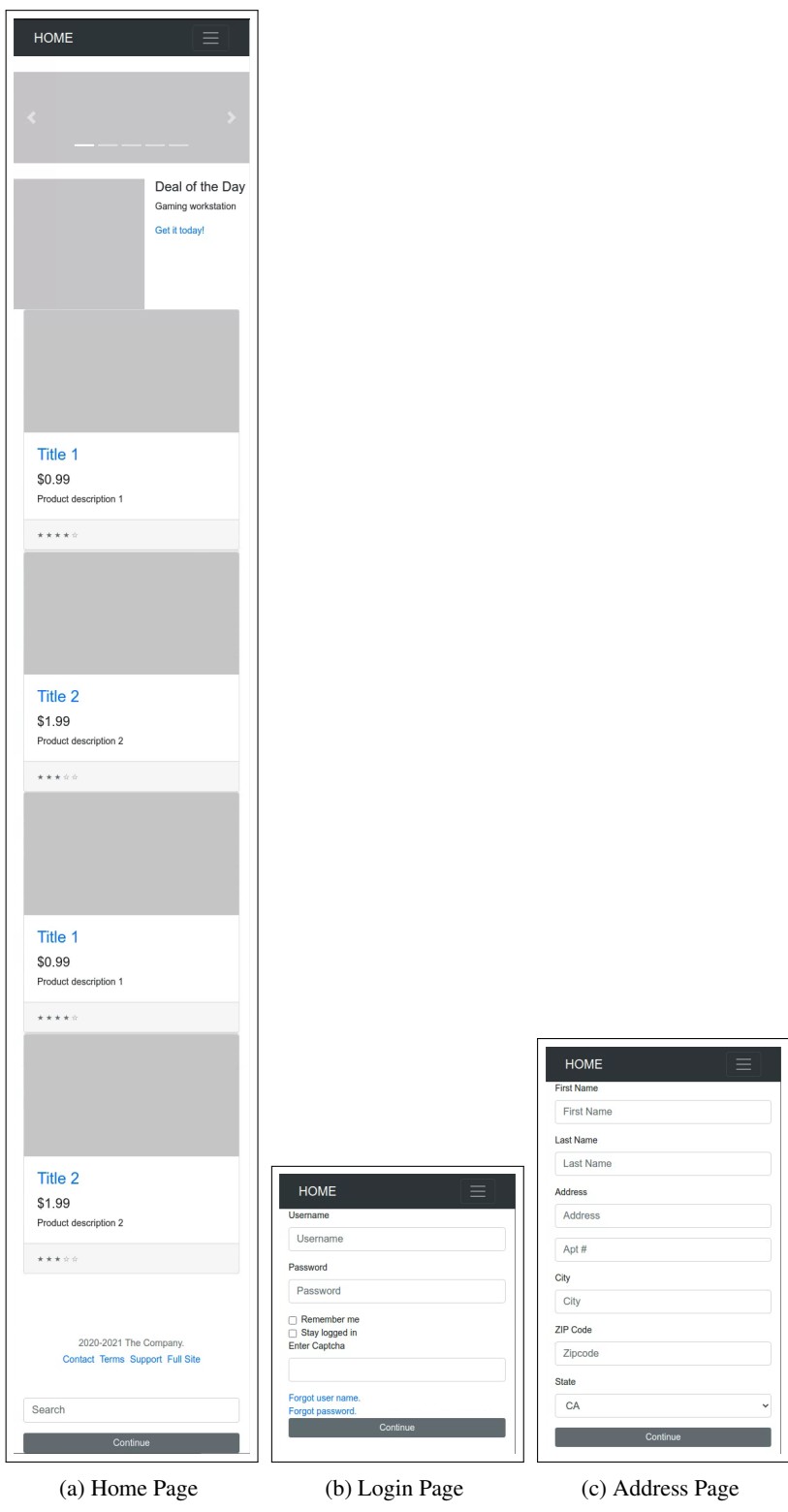

(a) Home Page       (b) Login Page       (c) Address Page

Figure 11: Screenshots of multi-page "Shopping" environment. The "Shopping" environment is composed of a complex home page and additional "Login" and "Enter Address" pages.

Figure 12: Screenshots of sample pages generated by the adversary.

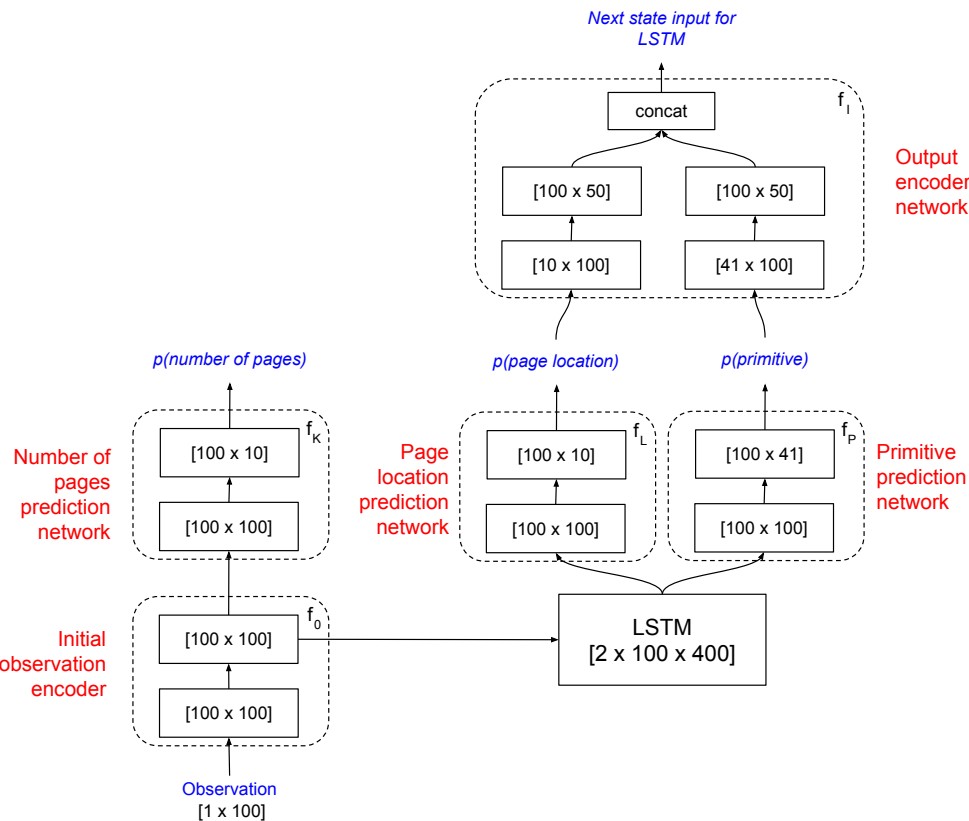

Figure 13: Each box corresponds to a dense layer with the shape of the corresponding linear transformation matrix. Each dense layer also includes a bias vector with the same size as the columns of the corresponding matrices.

