# OpenReview forum: "Adversarial Environment Generation for Learning  to Navigate the Web"
_ICLR.cc/2021/Conference — Reject_

### Official Review · AnonReviewer2 · 2020-10-27
**Good paper**

**Rating:** 7
**Confidence:** 3

**Review:**

This paper improves upon existing approaches for learning to fill forms on the web automatically. The main idea is to train an adversary to generate a curriculum of environments to train an agent to learn to fill forms on the web. Training such an adversary can be challenging since the adversary may prove to be too strong for the main agent to learn anything from. Thus, the paper proposes few techniques to control or shape this adversary such that the main agent is able to learn quickly as compared to similar existing approach.

The paper is well-written and clear for most parts. I feel the improvements the paper suggests are insightful and significant. The empirical analysis is also quite satisfactory and clearly shows the superiority of the proposed approach over existing baselines.

I dont have any major criticism for the paper as I quite like it. Here are a few minor points that I think the paper can improve on:
- The paper can benefit from a more general formulation of the task it solves. Currently, the paper is focused on web navigation tasks. This is not a weakness per se, but an abstract and general formulation of the task and solution can enhance the paper.
- On similar lines, the paper should consider toning down the claims a litte. Web navigation is quite a complex task that may use more than one aspect (if I may say so) of human intelligence. The paper presents an approach for matching provided information to the correct box in a web-form. This is quite simple as compared to web navigation.
- The paper can benefit from a few figures explaining the proposed system and experimental setup.

---

> ### Author Response · Authors · 2020-11-21
> **Revision in the paper writing with more experiments and figures**
>
> **"The paper can benefit from a more general formulation of the task it solves...."**
>
> Thank you for the suggestion. We agree that the core ideas of proposing an adversarial environment decoder that creates compositional tasks made with primitives, and training it on the objective that ties the adversary reward to the performance of the navigation agent and difficulty of the environment, are general beyond web navigation domain. Given that we only showed the method on the web navigation domain, we leave the formulation as it is in the paper.
>
> ---
> **"On similar lines, the paper should consider toning down the claims a litte..."**
>
> Thank you for your comment. We toned down the references to the real-world web navigation.  In addition, we revised the paper to be more specific about the multi-page form filling task and zero-shot transfer to new environments (Introduction) and highlight that the proposed method solves tasks an order of magnitude more complex than environments that state of the art in this domain.
>
> ---
>
> **"The paper can benefit from a few figures explaining the proposed system and experimental setup."**
>
> Thank you for the comment. We addressed this comment by adding Figure 6, restructuring the methods section, adding more details in the Algorithm 1, and adding Algorithm 2.

---

### Official Review · AnonReviewer3 · 2020-10-28
**The problem is interesting. The paper extends previous works and lacks details and results about their designed curriculum.**

**Rating:** 4
**Confidence:** 3

**Review:**

The paper studies the problem of learning to autonomously navigate the web, such as filling out web forms. It proposes a curriculum learning method that uses Adversarial Environment Generation (AEG) to build a curriculum of challenging web navigation tasks. It is based on the idea of creating training environments for RL agents. The paper is well written and easily accessible. The problem in this work is an interesting application of RL. The idea is not very new and extends from previous works using AEG.

It is difficult to compare the difference between the paper and PAIRED (Dennis et al., 2020). Because PAIRED (Dennis et al., 2020) is not public yet. However, as described in the related work, it seems like that the paper just applies AEG methods on complex, high-dimensional environments with some extension. The novelty of the work is limited.

In addition, the GAN part is also similar to Goal GAN (Florensa et al., 2017). The difference is how to label an environment (or a goal in Goal GAN). It is better to have more details about the adversary architecture. For example, how to label environments and train the adversary architecture.

It is unclear why the Adversarial Environment Generation (AEG) can provide a curriculum of learning different tasks. It is worth providing more experimental results about the progress of curriculum learning.

Overall, the main weaknesses are the novelty of the idea compared to previous works and how curriculum learning works for navigating the web tasks.

---

> ### Author Response · Authors · 2020-11-21
> **Revisions in the paper writing to clarify the novelty and more experiments**
>
> **"Overall, the main weaknesses are the novelty of the idea compared..."**
>
> Thank you for the comment. We hope that we addressed this question in the previous answers and the revision of the paper including the new experiments and visualizations.
>
> The key novelty in the domain of curriculum learning in web navigation is learned curriculum. The best of our knowledge this is the first work that applies a learned curriculum to the web navigation domain. To make that contribution, we present a new benchmarking environment and evaluation dataset (gMiniWob) and adversary architecture.
>
> In addition, the Flexible-bPAIRED advances the state of the art in the curriculum learning, proceeding a novel training technique that enables simultaneous control of the environment difficulty and the navigation agent’s competence. This is done through Eq (3). Objective Eq. (5) is novel and the empirical results show its significance in controlling the difficulty of the environments and proving the agents with just the right challenge.
>
> We are happy to discuss further if the reviewer had additional questions or wants the additional clarification.

---

> ### Author Response · Authors · 2020-11-21
> **Revisions in the paper writing to clarify the novelty and more experiments**
>
> **"It is difficult to compare the difference between the paper and PAIRED..."**
>
> Thank you for your feedback. We agree that the original submission did not sufficiently emphasise the novelty of the proposed method, and provided a good enough differentiation. We made a significant update in the paper’s Introduction, Related work, and Methods to highlight the contributions better. We emphasize that the changes made are only in writing, and the new version of the paper better describes the method. In addition PAIRED is now available at: https://papers.nips.cc/paper/2020/file/985e9a46e10005356bbaf194249f6856-Paper.pdf
>
> In summary, PAIRED proposes the idea of AEG trained with a regret reward, trains two navigation agents (antagonist and protagonist), and estimates the regret as the difference between antagonist and protagonist. There are two limitations with this work, centered around the sole dependence on regret as the difference between antagonist and protagonist. First, the adversary has no direct control over the difficulty of the environment and has no visibility into actual performance of the agents (how capable they are). Second, when the regret is zero, the adversary is forced to learn from sparse rewards. The regret is zero because a) both protagonist and antagonist are not collecting any reward, b) protagonist is more capable due to a more lucky initial seed, or c) both protagonist and antagonist are solving the environment equally well. All of these are less likely to appear in the simpler environments, like Mazes used in the original paper.
>
> We use the same framework, consisting of the adversary and population of the navigation agents, but our adversary training is very different. The most significant contribution is enabling the adversary to directly control the difficulty of the environment and closing the loop between the actual performance of the navigation agent, in addition to receiving the regret only. This results in the adversary being able to increase the difficulty of the environment when agents are capable and solve the presented environment, and reduce the difficulty when they struggle. We retain the regret reward which is used to select individual design elements, but compute it not from the fixed antagonist and protagonist agents, but flexibly as a difference between best performing and average performing agent in the population.
>
> The two policies, environment difficulty and individual design elements share the architecture and weights, and are trained via a decoder architecture (unlike PAIRED). The adversary learns both policies simultaneously, resulting in a hierarchical policy. To the best of our knowledge this training regiment is novel. Sections 4.2 and 4.3 are revised to clarify.
>
> The ability to adjust the environment difficulty based on the actual agent capability is a significant contribution, as evidenced in the performance of bPAIRED and Flexible-bPAIRED.
>
> ---
>
>
> **"In addition, the GAN part is also similar to Goal GAN (Florensa et al., 2017)..."**
>
> Thank you for the comment. We revised Section 4.2 to provide more details on the adversary and added a detailed comparison to Goal GAN (Florensa et al., 2017). The adversary is an environment decoder, consisting of a seed drawn from a random standard normal distribution, LSTM with two FF nets that output elements and its location. The adversary is rolled out in an open loop fashion and trained without the ground truth. In summary, Goal GAN requires bootstrapping the generator from sample goals that the initial agent is able to solve (initialize_GAN), only evaluated on simple navigation tasks (similar to PAIRED (Dennis et al., 2020), and assumes fixed environment dynamics that is shared between training and testing. In contrast, we start with empty environments in which bootstrapping a generator from episodes is not possible, test on more complicated high-dimensional state and action spaces, train an adversary that generates evolving environment dynamics, and test on unseen environments.
>
> ---
>
> **"It is unclear why the Adversarial Environment Generation (AEG) ..."**
>
> The key mechanism for the curriculum is the addition of the loss depicted in Eq. (5) which ties in the adversary objective directly to the performance of the agent when selecting the environment difficulty.
> We added more experiments to detail the average number of active and passive elements generated during training in the Appendix (Figure 8). We show that both numbers start small and increase over time with different characteristics that results in gradually more complex websites while web navigation agents are still learning better policies (Figure 4 and 5).

---

### Official Review · AnonReviewer4 · 2020-10-28
**Generating Environments through Flexible-(b)PAIRED Algorithms**

**Rating:** 5
**Confidence:** 3

**Review:**

The main research contribution of this paper is the Flexible-PAIRED, and bPAIRED algorithms for generating Web Sites that can be then "used" by automatic agents. The modifications to the original PAIRED algorithm is quite simple but proven to be effective in the paper.

Honestly, I believe the paper has some merit but in the current form is quite difficult to be understood.

First and foremost, it is really not clear what are the motivations behind the research work. I understand that you want to learn a policy for filling in a web form but, honestly, I am not sure I got that. The examples you make are not super clear, you should give an idea, for instance, of what's the complexity of the tasks in the examples. For instance, booking a flight to LAX for Friday involves: i) inputting LAX in an entry box, clicking search button, select flights according to user preferences (how are these taken into account, BTW?) and then pay using a credit card. Now, all of these passages are quite difficult and complex and I am not sure the technique presented in the paper actually can do that. From the paper it is not clear what are the tasks considered and how they are generated.

The second question I have regards the relevance of the results. While the difference between PAIRED and Flexible-PAIRED and Flexible-bPAIRED are large, it is not clear that a success rate of 80% can be considered high enough for real systems.

One thing that should be improved is readability, in general. For instance, the description of the PAIRED algorithm is quite difficult to follow. I understand that this is not an algorithm that is developed by the authors but the description should be done in a correct way that will make the reader understand what are the shortcomings and why you had to propose Flexible-PAIRED and bPAIRED.

A relatively minor point that needs explanation regards the impact of the simplification of the generation process described in Section 4. It is not clear if the limitations of the primitives used also limit greatly the diversity of the generated pages making the task simpler.

It is true, though, that results are much better than the baseline and for this reason I believe the paper has some merit.

Minor issues:
1. The first sentence in the Introduction is difficult to understand and does not set correctly the goals of the paper. What do you learn to generate? DOMs or Tasks + DOMs?
2. The regret will be highest for easy --> the regret will be higher
3. "assume links between pages are implicitly defined by events attached to certain elements." --> This sentence is not clear

---

> ### Author Response · Authors · 2020-11-21
> **Revisions in the paper writing for clarity, readability, and better comparison**
>
> **"A relatively minor point that needs explanation regards..."**
>
> It is true that if we generate a website from a limited set of primitives, not arbitrary DOM generation processes, it will be limited. However, this doesn’t necessarily mean simpler tasks as the order of the primitives in the DOM significantly changes the observation space, making it much difficult for webnav agents to learn. This is a combinatorial problem. For example, consider just using two possible design templates: input field and submit button, and creating a three page task where each web page constraints 10 different input fields that need to be correctly filled in then submit button clicked, to proceed to the next page. The full task is completed only after all the pages. This task completion requires correctly filling in all 30 fill in boxes, and the submit buttons on the individual pages only after all the fill in boxes are completed. This is a rather challenging task even though we use only two design templates.
>
> An equivalent problem in maze navigation would be a game, where the agent needs to complete a series of levels. On each level it needs to correctly manipulate and complete a number of sub tasks (fill in boxed, drop downs etc.) to unlock a door (submit button), before finding the door to proceed to the next level. In the web navigation evaluation, we count tasks successful only after all phases are complete without giving partial-credit. That said, during the training, we do reward the agent completing a sub task.
>
> Also, generating a DOM from scratch is not feasible as it might produce inconsistent websites. Consider an example where at one iteration the adversary adds the “First Name” label while in the next iteration it adds the “Last Name” label. This causes the web navigation agents to train on inconsistent website samples. Using the comparison with the maze navigation, generating DOM from scratch would be an equivalent of creating Minecraft or Atari game environments from pixels, instead of selecting semantic objects.
>
> We clarify these points in Section 4.1.
>
> ---
>
> **"Minor issues:..."**
>
> Thank you for the feedback. We have significantly revised the instruction to focus on the form-filling web navigation problem and learning to generalize these tasks across unseen environments. We learn to generate environments for the form filling tasks, which consist of the set of web pages, and an instruction that defines the task.

---

> ### Author Response · Authors · 2020-11-21
> **Revisions in the paper writing for clarity, readability, and better comparison**
>
> **"First and foremost, it is really not clear what are the motivations behind the research work..."**
>
> Thank you very much for the feedback. We have significantly revised the manuscript to clarify the problem and complexity that we are addressing. To that end:
>
> * In the introduction, we added the first two paragraphs that describe the task and the complexity of it.
> * We added Table 2 that explicitly presents the complexity of the proposed new benchmark over the existing one, showing an order of magnitude more complex web pages in terms of number of elements, instruction size, three depth size.
> * gMiniWob contains a multi-page navigation environment (shopping) which consists of three pages: home page, log in, and address. The existing benchmark contains only a single page form filling tasks.
> * We improved the description of the proposed benchmark.
>
> ---
>
> **"The second question I have regards the relevance of the results..."**
>
> Thank you for the feedback. The reviewer is correct that 80% is not sufficient for the real, production ready web navigation. That is not our goal. In this paper, we aim to draw attention to the web navigation and a challenging and compelling research problem. To that end, we contribute both a method that improves the state of the art, and a benchmarking environment (to be open sourced) to encourage future research and contributions in this space. We have revised the manuscript to reduce the emphasis on the real-world navigation, motivate better web navigation (and form filling in particular) as a compelling research topic (in the introduction), emphasized the benchmarking environment as a contribution, and revised Section 4.1 to describe the environment in more detail.
>
> ---
>
> **"One thing that should be improved is readability, in general..."**
>
> Thank you very much for this comment. We made a detailed pass through the manuscript and improved readability throughout. Specifically:
>
> * We restructured the introduction to better motivate our contributions. Fourth paragraph on page 2 outlines PAIRED short cominging, and the next paragraph describes contributions.
> * Related works section contains better contrast with PAIRED.
> * To simplify the exposition, we only present Flexible-bPAIRED as a contribution, and use bPAIRED and Flexible-PAIRED as ablation methods.
> * We made a detailed pass through the methods section to further separate it from PAIRED and present only the contributions that are novel to this work. This includes adding more details that were previously omitted. Specifically, the original submission did not emphasize the adversary architecture and the training losses sufficiently.
>
> We want to clarify that we did not change the method in this revision -- only revised the writing to better reflect the methods we are presenting. In particular the budget enforcing mechanism and training adversary method by simultaneously training two policies with shared weights: difficulty selection and actual design elements, are to the best of our knowledge unique and novel. And the empirical results, seen in the strength of the bPaired and Flexible-bPAIRED provides the empirical evidence for its significance.

---

### Official Review · AnonReviewer1 · 2020-10-30
**Interesting application, seemingly solid work**

**Rating:** 6
**Confidence:** 1

**Review:**

This paper presents a technique for adversarial generation of environments for the interesting problem of web navigation, and provides an environment that enables learning to design complex websites out of a set of compositional primitives. Then, it also proposes a method to adversarially generate a curriculum of increasingly complicated websites, and uses it to train agents which can navigate more challenging, high-dimensional websites.

Strengths:
1. An interesting novel problem domain; which is going to be very useful in a number of human computer interaction applications.
2. The web navigation environment is interesting - it will hopefully spur more research for this problem
3. The training of agents with an autoregressive adversary policy is interesting.

Weaknesses:
1. The discussion on related work on this application seems sparse; hence, for me, it was hard to judge the novelty of this work.
2. More discussion of the environment - some examples, what makes it hard, or easy would help the reader understand the key challenges.
3. More discussion on past work in interactive learning with autoregressive adversarial policies would be helpful. It will help the reader understand why this is a different interactive task and what makes it more interesting or challenging.
4. The experimental section is too sparse. Some more ablation studies on different parts of the model - e.g. budget enforcing on the adversary would be helpful.
5. The b-paired and flexible b-paired agents seem to be very similar to each other - especially for some problems. Some more analysis of this would be useful.

---

> ### Author Response · Authors · 2020-11-21
> **Revisions in the paper writing and new experiments**
>
> 1. Thank you for the constructive feedback. We have significantly revised the introduction and related work sections.
>
>
> 2. To explain better the difficulty of the environments, we added:
>   * The second paragraph in the Introduction to describe the difficulty of the tasks and environment.
>   * Added in Appendix additional images from the generated environments, and discussed the size of the adversary and web navigation agents.
>
>
> 3. We added more discussions on the autoregressive architecture of our model and its comparison to other autoregressive models in the related work section.
>
>
> 4. Thank you for this feedback. We added new experimental results with different budget weights within {0.25, 0.5, 0.75, 1.0, 1.25} in the Appendix. In summary, a small budget weight (0.25) gives more importance to the RL loss in Eq. (3) with a substantial drop in performance, signifying the emphasis of the proposed budget weighting. But, Flexible-bPAIRED is still able to outperform other models in all settings.
>
>
> 5. Thank you for the feedback. In the revised version of the paper we retain flexible b-paired as a contribution and use b-paired as an ablation method. The rest of the paper -- introduction, methods, and results is updated to this effect.  Flexible b-paired offers a performance edge after training longer indicating that both flexible agent selection and budget mechanism are complementary (Figure 4). We observe that the budget mechanism always encourages the adversary to design challenging but solvable environments while flexible agent selection improves the regret estimation to be positive.

---

### Decision · Program_Chairs · 2021-01-07
**Final Decision**

**Decision:**

Reject

**Comment:**

This paper considers the problem of agents learning to autonomously navigate the web, specifically by focusing on filling out forms. The focus is on using adversarial environment generation to form a curriculum of training tasks.
Thank you for the revisions to the manuscript, which have particularly improved readability.
The presented problem is really interesting and seems an important real-world problem for RL.
Despite this, as the paper stands the results are not completely convincing. It seems like there is also scope to rigourously analyse the proposed method on other, better known domains to better quantify its limitations.